# ATPase Inhibitory Factor-1 Disrupts Mitochondrial Ca^2+^ Handling and Promotes Pathological Cardiac Hypertrophy through CaMKIIδ

**DOI:** 10.3390/ijms22094427

**Published:** 2021-04-23

**Authors:** Mario G. Pavez-Giani, Pablo I. Sánchez-Aguilera, Nils Bomer, Shigeki Miyamoto, Harmen G. Booij, Paula Giraldo, Silke U. Oberdorf-Maass, Kirsten T. Nijholt, Salva R. Yurista, Hendrik Milting, Peter van der Meer, Rudolf A. de Boer, Joan Heller Brown, Herman W. H. Sillje, B. Daan Westenbrink

**Affiliations:** 1Department of Cardiology, University Medical Center Groningen, University of Groningen, P.O. Box 30.001, 9700 RB Groningen, The Netherlands; mario.pavez@med.uni-goettingen.de (M.G.P.-G.); p.i.sanchez.aguilera@umcg.nl (P.I.S.-A.); n.bomer@umcg.nl (N.B.); hg.booij@umcg.nl (H.G.B.); giraldopaula01@gmail.com (P.G.); s.u.oberdorf@umcg.nl (S.U.O.-M.); k.t.nijholt@umcg.nl (K.T.N.); SYURISTA@mgh.harvard.edu (S.R.Y.); p.van.der.meer@umcg.nl (P.v.d.M.); r.a.de.boer@umcg.nl (R.A.d.B.); h.h.w.sillje@umcg.nl (H.W.H.S.); 2Department of Pharmacology, University of California San Diego, San Diego, CA 92093, USA; smiyamoto@ucsd.edu (S.M.); jhbrown@health.ucsd.edu (J.H.B.); 3Erich and Hanna Klessmann Institute, Heart and Diabetes Center NRW, University Hospital of the Ruhr-University Bochum, Georgstrasse 11, 32545 Bad Oeynhausen, Germany; hmilting@hdz-nrw.de

**Keywords:** mitochondria, calcium handling, heart failure, CaMKII, cardiomyocyte hypertrophy

## Abstract

ATPase inhibitory factor-1 (IF1) preserves cellular ATP under conditions of respiratory collapse, yet the function of IF1 under normal respiring conditions is unresolved. We tested the hypothesis that IF1 promotes mitochondrial dysfunction and pathological cardiomyocyte hypertrophy in the context of heart failure (HF). Methods and results: Cardiac expression of IF1 was increased in mice and in humans with HF, downstream of neurohumoral signaling pathways and in patterns that resembled the fetal-like gene program. Adenoviral expression of wild-type IF1 in primary cardiomyocytes resulted in pathological hypertrophy and metabolic remodeling as evidenced by enhanced mitochondrial oxidative stress, reduced mitochondrial respiratory capacity, and the augmentation of extramitochondrial glycolysis. Similar perturbations were observed with an IF1 mutant incapable of binding to ATP synthase (E55A mutation), an indication that these effects occurred independent of binding to ATP synthase. Instead, IF1 promoted mitochondrial fragmentation and compromised mitochondrial Ca^2+^ handling, which resulted in sarcoplasmic reticulum Ca^2+^ overloading. The effects of IF1 on Ca^2+^ handling were associated with the cytosolic activation of calcium–calmodulin kinase II (CaMKII) and inhibition of CaMKII or co-expression of catalytically dead CaMKIIδC was sufficient to prevent IF1 induced pathological hypertrophy. Conclusions: IF1 represents a novel member of the fetal-like gene program that contributes to mitochondrial dysfunction and pathological cardiac remodeling in HF. Furthermore, we present evidence for a novel, ATP-synthase-independent, role for IF1 in mitochondrial Ca^2+^ handling and mitochondrial-to-nuclear crosstalk involving CaMKII.

## 1. Introduction

The heart requires tremendous amounts of ATP to sustain the systemic circulation, most of which is generated by mitochondria. In patients with heart failure (HF), metabolic roadblocks and structural damage to mitochondria occur that induce mitochondrial oxidative stress and diminish myocardial ATP production [1]. It has become increasingly clear that “retrograde” signals originating from dysfunctional mitochondria activate nuclear gene transcription that promotes maladaptive cardiac growth [2]. Enhanced understanding of the pathways underlying this retrograde mitochondrial-to-nuclear crosstalk in HF could uncover new treatments for HF patients.

ATP synthase is a reversible rotary enzyme that synthesizes ATP when driven by a mitochondrial membrane potential (Δψ_m_) that is generated by the respiratory chain [3]. When Δψ_m_ dissipates, under ischemic conditions or in critically damaged mitochondria, ATP synthase reverses its direction of rotation and becomes an ATP consumer rather than a generator. Reversal of ATP synthase is specifically blocked by the evolutionary conserved mitochondrial protein ATPase inhibitory factor-1 (IF1 or ATPIF1) [4]. By blocking ATP synthase reversal, IF1 conserves ATP and permits eradication of depolarized dysfunctional mitochondria; thereby preserving cellular viability under ischemic conditions [5].

It has long been thought that the biological function of IF1 was restricted to conditions of respiratory collapse. Recent studies have, however, uncovered important physiological roles for IF1 under respiring conditions, including, but not restricted to, the regulation of oxidative stress [6], Ca^2+^ handling [7], cristae formation [5,7,8,9], glucose metabolism [10,11], and neuronal development [12,13]. While the role of binding to ATP synthase in these processes is disputed, there is general consensus that many of these non-canonical roles of IF1 are governed by mitochondrial-to-nuclear signaling [5,6,7,10,11,14].

Yang et al. were recently the first to demonstrated that the expression of IF1 is increased in HF and that IF1 knockout mice are less susceptible to pressure overload-induced HF [14]. Based on the above, we hypothesized that IF1 could represent a critical regulator of maladaptive mitochondrial-to-nuclear crosstalk in HF.

In the present study, we discovered a key role for IF1 in mitochondrial reprograming and maladaptive remodeling in cardiomyocytes that is independent of binding to ATP synthase. Instead, we uncovered a novel and translationally relevant pathway that positions IF1 at the intersection of mitochondrial Ca^2+^ handling and mitochondrial-to-nuclear crosstalk in cardiomyocytes. By doing so, we provide direct evidence for a physiological role for IF1 that is truly independent of ATP synthase.

## 2. Results

### 2.1. Myocardial Expression of IF1 Is Increased in Heart Failure and Induces Metabolic Reprogramming in Neonatal Rat Ventricular Myocytes (NRVMs)

IF1 mRNA expression was increased in the left ventricle (LV)lysates obtained from several models of murine HF; including mice with cardiomyocyte autonomous expression of activated Gαq [15], mice subjected to transverse aortic constriction (TAC) [16], or mice subjected to a myocardial infarction (MI) [16] (Figure 1A). The increased mRNA expression of IF1 corresponded with a similar increase in IF1 protein levels (Appendix A). IF1 mRNA levels were also higher in cardiac lysates obtained from patients with end-stage HF compared to those in lysates from control hearts (Figure 1B). In neonatal rat ventricular myocytes (NRVMs), overexpression of Gαq (Appendix A) and treatment with isoproterenol (Figure 1C), which both stimulate pathological hypertrophy resulted in a significant increase in the expression of IF1. In contrast, treatment with insulin-like growth factor-1 (IGF-1), which stimulates physiological cardiac growth did not affect IF1 expression levels (Figure 1C). The expression of IF1 increased during consecutive stages of cardiac development and declined after birth (Appendix A), suggesting that IF1 represents a member of the fetal-like gene program that is activated downstream of neurohumoral signaling pathways.

Next, NRVMs were infected with an adenoviral vector expressing human IF1 (ad-IF1-WT) or a control virus expressing GFP (ad-CTL). Infection with ad-IF1-WT increased IF1 protein levels by sevenfold (Appendix A), which resulted in a stronger inhibition of the reverse mode of ATP synthase (Appendix A) and the preservation of intracellular ATP levels after respiratory collapse (Appendix A). Infection with ad-IF1-WT did not influence cellular viability (Appendix A).

IF1 did not influence basal or ATP-linked respiration nor did it affect the mitochondrial proton leak (Figure 1D,E and Appendix A). IF1 expression did, however, result in a dose-dependent reduction in the maximal mitochondrial respiration and the mitochondrial spare capacity (Figure 1E and Appendix A). IF1 knockdown with siRNA did not influence mitochondrial respiration, arguing against a direct effect of IF1 on respiratory complexes (Appendix A). Next, we determined the consequences of IF1 on extramitochondrial metabolism by measuring extracellular acidification rates as a proxy of anaerobic glycolysis. IF1 also increased in anaerobic glycolysis (Figure 1F,G), accompanied by increased expression of the key glycolytic enzymes lactate dehydrogenase (*LDH*) and pyruvate kinase (*PRK*) (Figure 1H). Together, these findings suggest that IF1 compromises maximal mitochondrial respiration and promotes extramitochondrial glycolysis without affecting ATP-linked respiration.

### 2.2. IF1 Induces Mitochondrial Oxidative Stress and Stimulates Cardiomyocyte Hypertrophy

IF1 has been shown to both increase or decrease mitochondrial reactive oxygen species (ROS) emissions depending on the cell type and experimental context [5,9,17]. Infection with IF1 resulted in a 40% increase in fluorescence intensity of the mitochondrial-specific ROS indicator MitoSOX (Figure 2A). Expression of the ROS-responsive genes NADPH oxidase 2 (*NOX2*), nuclear receptor factor-2 (*NRF2*), and heat shock protein-60 (*HSP60*) were also significantly increased by IF1 (Figure 2B) as was ROS-mediated damage to mitochondrial DNA (Figure 2C). In addition, the protein expression of four out of five respiratory chain complexes was significantly reduced in ad-IF1-WT infected cells (Figure 2D). The reductions in maximal mitochondrial respiration induced by IF1 are, thus, likely explained by mitochondrial oxidative stress and reductions in electron chain complexes.

Overexpression of IF1 also resulted in a marked cardiomyocyte hypertrophy (Figure 2E), accompanied by increases in mRNA levels of atrial natriuretic peptide (*ANP*) and B-type natriuretic peptide (*BNP*) (Figure 2F). The increases in cell size and natriuretic peptides were equivalent to those of 50 µM of phenylephrine (Appendix A). The combination of phenylephrine and ad-WT-IF1 did not result in further increase in cell size or ANP (Appendix A). IF1 expression, thus, appears to be sufficient to promote pathological cardiomyocyte hypertrophy.

### 2.3. IF1-Induced Cardiomyocyte Reprograming Occurs Independent of Binding to ATP Synthase

To determine whether the effects of IF1 observed in our model were dependent on the binding of IF1 to ATP synthase, part of the experiments in Figure 1 and Figure 2 were repeated with an adenovirus expressing a mutant form of IF1 in which glutamic acid 55 had been replaced by alanine (ad-IF1-E55A). This mutation renders IF1 incapable of binding to ATP synthase [18]. Infection with ad-IF1-E55A resulted in similar increases in IF1 expression compared to that for ad-IF1-WT (Appendix A). As expected, the drop in mitochondrial membrane potential following respiratory collapse was reduced by ad-IF1-E55A, consistent with diminished inhibition of the reverse mode of ATP synthase (Appendix A).

Despite its inability to bind to ATP synthase, ad-IF1-E55A induced similar reductions in maximal mitochondrial respiration and mitochondrial spare capacity as ad-WT-IF1 (Figure 3A,B). Infection with ad-IF1-E55A also resulted in a marked increase in glycolysis (Figure 3C,D) as well as increased expression of *LDH* and *PRK* (Figure 3E). Mitochondrial ROS emissions were also increased after infection with ad-IF1-E55A (Figure 3F), as were the expression levels of *NOX2*, *NRF2*, and *HSP60* (Figure 3G). Moreover, downregulation of the respiratory chain complexes II, IV, and V was also observed with ad-IF1-E55A (Appendix A). 

Finally, infection with ad-IF1-E55A induced cardiomyocyte hypertrophy that was proportional to what had been observed with ad-IF1-WT (Figure 3H), and *ANP* and *BNP* mRNA expression were also significantly increased (Appendix A). Together, these data demonstrate that the effects of IF1 on mitochondrial function and cardiomyocyte hypertrophy are independent of the interaction between IF1 and ATP synthase.

### 2.4. IF1 Induces Mitochondrial Fission through Dynamin-Related Protein 1 (DRP1)

IF1 regulates mitochondrial dynamics, and perturbations in mitochondrial dynamics are thought to contribute to mitochondrial dysfunction in heart failure [19,20,21,22]. To assess whether IF1 altered mitochondrial dynamics, mitochondrial volume and morphology were assessed using confocal Z/y scanning followed by image deconvolution and 3D reconstruction. Typical examples of confocal images and multislice 3D reconstructions are depicted in Figure 4A,B. Total mitochondrial volume and mitochondrial DNA copy numbers were not affected by IF1(Appendix A). Infection with ad-IF1-WT did, however, increase the number of mitochondria per cell and reduced the average mitochondrial volume to increase the fission index (Figure 4C). Mitochondrial fission is governed by translocation of the GTPase protein called dynamin-related protein 1 (DRP-1) to the outer mitochondrial membrane [23]. Consistent with the results from the imaging analysis, infection with IF1 resulted in a marked increase in the mitochondrial translocation of DRP1 (Figure 4D,E). In addition, the myocardial expression of mitochondrial fusion protein, mitofusin-2, was decreased by IF1 (Figure 4F,G). Under the current conditions, the mitophagic flux was not affected by IF1 as we did not observe mitochondrial translocation of Parkin or changes in the protein level of PTEN-induced kinase 1 (PINK1) (Figure 4D–G). In summary, we demonstrate that IF1 overexpression promotes mitochondrial fission without increasing mitophagic flux.

### 2.5. IF1 Reduces Mitochondrial Ca^2+^ and Induces Sarcoplasmic Reticulum Ca^2+^ Overload

Excessive mitochondrial fission reduces the mitochondrial capacity to store Ca^2+^ and contributes to mitochondrial dysfunction in heart failure [20,24,25]. We, therefore, sought to determine whether IF1 affects mitochondrial and cellular Ca^2+^ homeostasis. Infection with ad-IF1-WT resulted in a significant reduction in mitochondrial Ca^2+^ concentrations detected with the mitochondrial Ca^2+^ indicator Rhod2-AM (Figure 5A). Similarly, FCCP-induced mitochondrial Ca^2+^ release was also significantly reduced by IF1 (Figure 5B,C).

We next determined how the reductions in mitochondrial Ca^2+^ content affected Ca^2+^ concentrations in other cellular compartments. Expression of ad-IF1-WT did not affect basal cytosolic Ca^2+^ levels (Figure 5D). However, sarcoplasmic Ca^2+^ content was significantly increased by IF1 (Figure 5E,F). To explore the potential molecular mechanisms responsible, the mRNA expression of a panel of genes involved in the regulation of mitochondrial Ca^2+^ handling was assessed. Interestingly, IF1 resulted in a significant increase in the expression of the dominant-negative pore-forming subunit of the mitochondrial Ca^2+^ uniporter ß (*MCUB*, Figure 5G). Together, these data indicate that IF1-induced mitochondrial fragmentation and upregulation of MCUB compromise mitochondrial Ca^2+^ handling and induce sarcoplasmic reticulum (SR) Ca^2+^ overloading.

### 2.6. IF1-Induced Cardiomyocyte Hypertrophy Is Dependent upon CaMKIIδ

SR Ca^2+^ overloading represents a critical step in the development of pathological cardiomyocyte hypertrophy as it activates maladaptive Ca^2+^-sensitive signaling pathways such as Ca^+2^/calmodulin protein kinase II (CaMKII) and calcineurin [26,27,28]. To determine the effect of IF1 on CaMKII activation, we measured CaMKII autophosphorylation (Thr286) as well as the phosphorylation of the CaMKII-specific sites on its downstream targets phospholamban (Thr17) and adenosine monophosphate-activated protein kinase (AMPK; Thr172). Infection with ad-IF1-WT resulted in CaMKII autophosphorylation and marked increases in the phosphorylation of its downstream targets (Figure 6A,B). Protein kinase A–dependent phosphorylation of CaMKII (Thr16) was not increased by IF1 (Appendix A). No changes in the mRNA expression of calcineurin response element 1 (RCAN1) were detected, suggesting that calcineurin was not activated by IF1 (Appendix A). Infection with ad-IF1-E55A had similar effects on mitochondrial Ca^2+^ handling and the activation of CaMKII-signaling (Appendix A), reinforcing the notion that these properties of IF1 are independent of its binding to ATP synthase.

Next, NRVMs were co-infected with an adenovirus expressing a catalytically dead mutant of the predominant cardiac isoform of CaMKII (CaMKIIδ; ad-dnCAMKII). As expected, infection with ad-dnCAMKII reduced CaMKII-dependent phosphorylation of phospholamban (PLN) (Figure 6C). Co-infection with ad-dnCAMKII did not alter IF1-induced changes in maximal mitochondrial respiration (Figure 6D) nor did it affect mitochondrial ROS production (Figure 6E). Ad-dnCAMKII did, however, block IF1-induced cardiomyocyte hypertrophy (Figure 6E) and IF1-induced expression of *ANP* (Figure 6F). Together, these findings indicate that the effects of IF1 on cardiac hypertrophy are dependent on the activation of CaMKIIδ (Figure 7).

## 3. Discussion

In the present study we investigated the role of IF1 in mitochondrial function and cardiomyocyte remodeling. We showed that the myocardial expression of IF1 was increased in mice and in humans with HF, downstream of neurohumoral signaling pathways, and in patterns that resemble the fetal-like gene program. Expression of IF1 in NRVMs induced metabolic changes that resembled the failing heart, including reductions in mitochondrial respiratory capacity, mitochondrial oxidative stress, and enhanced extramitochondrial glycolysis. Expression of IF1 alone was sufficient to induce pathological cardiomyocyte hypertrophy, which was found to occur independent of binding to ATP synthase. Instead, IF1compromized mitochondrial calcium handling through excessive mitochondrial fission and upregulation of *MCUB*. Together, this resulted in reductions in mitochondrial Ca^2+^, SR Ca^2+^ overloading, and the cytosolic activation of CaMKIIδ. Finally, IF1-mediated cardiomyocyte hypertrophy was found to be depend on the activation of CaMKIIδ. Our study has uncovered IF1 as a novel member of the fetal-like gene program that contributes to mitochondrial dysfunction and pathological cardiac remodeling in HF. Furthermore, we present evidence for a novel role for IF1 in mitochondrial Ca^2+^ handling and mitochondrial-to-nuclear crosstalk involving CaMKIIδ.

The primary function of IF1 is to block the reverse mode of ATP synthase, which can occur upon dissipation of the mitochondrial membrane potential under ischemic conditions or in critically damaged mitochondria [14]. By blocking ATP synthase reversal, IF1 prevents mitochondria from becoming ATP consumers rather than generators. Under these conditions, IF1 also suppresses programmed cell death and stimulates Parkin-dependent elimination of dysfunctional mitochondria [8,9,22]. It has long been thought that the biological function of IF1 was restricted to conditions of respiratory compromise. Several recent publications have, however, suggested that IF1 can also inhibit the forward direction of ATP synthase and, thereby, control the rate of ATP synthesis under normal respiring conditions [6]. Conversely, different research groups have rather demonstrated that IF1 promotes ATP synthase activity by stimulating mitochondrial cristae formation [8,9]. These conflicting results may reflect differences in cell types and experimental conditions, as most studies so far have been performed in immortalized cell lines and cancer models. In our hands, IF1 did not influence ATP-linked respiration and the structural and biochemical consequences of overexpression of IF1 also occurred with an IF1 mutant that was unable to bind to ATP synthase. This strongly suggest that the biological effects of IF1 under normal respiring conditions are independent of the canonical pathway that requires binding to ATP synthase.

Overexpression of IF1 reduced maximal mitochondrial respiration and stimulated glycolysis. The reductions in mitochondrial respiratory capacity were associated with mitochondrial oxidative stress, significant damage to mitochondrial DNA, and downregulation of mitochondrial respiratory chain complexes. In addition, IF1 stimulated mitochondrial fission and fragmentation, which by itself is sufficient to reduce mitochondrial respiration and promote oxidative stress [29]. IF1 has consistently been implicated in mitochondrial oxidative stress, yet the exact mechanism responsible remains enigmatic. Some authors have suggested that IF1-induced oxidative stress is caused by increases in the mitochondrial membrane potential, secondary to ATP synthase inhibition [5]. However, our results clearly indicate that the mitochondrial oxidative stress induced by IF1 is independent of its capacity to bind to ATP synthase. Further research is, therefore, required to determine the mechanisms of IF1-induced mitochondrial ROS.

The profound effects of IF1 on mitochondrial Ca^2+^ handling were arguably the most intriguing finding of our study. Overexpression of IF1 reduced the mitochondrial capacity to store Ca^2+^ and resulted in SR Ca^2+^ overloading. Aberrant SR Ca^2+^ handling is a central mechanism responsible for various pathophysiological changes in failing myocytes including, but not restricted, to cardiac arrythmias, transcriptional activation of hypertrophy, mitochondrial dysfunction, and cell death [30]. IF1 could, thus, reflect a novel mechanistic link between mitochondrial dysfunction and dysregulated Ca^2+^ handling in HF. Our findings confirm and extend upon a recent study in which exogenous treatment with IF1 corresponded with increases in cytosolic Ca^2+^ levels in skeletal muscle cells [11]. The mechanisms responsible for IF1-induced reductions in mitochondrial Ca^2+^ are unknown, yet it is tempting to speculate that they rely on intrinsic biochemical properties of the IF1 molecule. For instance, it was recently discovered that Ca^2+^ influences the self-association and structure of IF1, and it is possible that IF1 influences the Ca^2+^ buffering in the mitochondrial matrix [31]. IF1 has also been shown to regulate critical Ca^2+^-handling proteins such as the MCU [7]. While we did not detect changes in the expression of MCU, IF1 did promote the expression of the negative regulator of MCU, MCUB. Finally, we cannot exclude the possibility that the reductions in mitochondrial Ca^2+^ were the consequence of mitochondrial oxidative stress or the mitochondrial fragmentation observed in our model. Nevertheless, our study contributes to the growing body of evidence that IF1 regulates mitochondrial Ca^2+^.

IF1-induced SR Ca^2+^ overloading was associated with the activation of CaMKII and CaMKII-dependent induction of cardiomyocyte hypertrophy. Our results are in line with studies in global IF1 knockout mice, which appeared to be protected from pressure overload-induced cardiomyocyte hypertrophy [14]. We confirm and extend upon these observations by providing a mechanistic link between SR Ca^2+^ overloading and the activation of CaMKIIδ. Multifunctional CaMKII has been implicated in a myriad of pathogenic cellular responses in HF, which include mitochondrial reprograming, mitochondrial oxidative stress, and mitochondrial fragmentation [15,27,32,33]. Nevertheless, IF1-induced reductions in mitochondrial respiration and mitochondrial ROS were not affected by catalytically dead CaMKIIδ. The activation of CaMKIIδ, thus, appears to be a consequence of IF1-induced mitochondrial dysfunction rather than a cause. Of note, CaMKIIδ can also be activated by ROS, and we cannot determine whether the activation of CaMKIIδ in our model was dependent upon Ca^2+^ or ROS.

While our study provides compelling evidence for a role of IF1 in controlling mitochondrial oxidative stress and mitochondrial Ca^2+^ homeostasis in cardiomyocytes, several limitations need to be acknowledged. One of the main limitations of our experimental model is that neonatal cardiomyocytes display a relatively immature metabolic phenotype that is highly glycolytic. Therefore, the role of IF1 on mitochondrial metabolism may be different in adult cardiomyocytes or in vivo models. Moreover, ATP-linked respiration is modest in NRVMs, which may have prevented us from detecting subtle changes in ATP-linked respirations induced by IF1. Nevertheless, studies with the ad-IF1-E55A mutant clearly demonstrated that neither the metabolic nor the structural changes induced by IF1 were dependent on the binding of IF1 to ATP synthase. Another limitation of our study is that we do not provide mechanisms responsible for IF1-induced mitochondrial oxidative stress. In addition, we do not provide mechanistic insights into the role of mitochondrial oxidative stress in cardiomyocyte hypertrophy downstream of IF1. In our opinion, this is beyond the scope of the current investigation, and future studies are required to address these questions.

Based on our results, we suggest that the increase in the expression of IF1 in HF is maladaptive and contributes to mitochondrial fragmentation, aberrant Ca^2+^ handling, and CAMKIIδ-dependent pathological remodeling. Our study reinforces the concept that mitochondrial dysfunction has profound effects in failing cardiomyocytes that extend beyond bioenergetic insufficiency. Finally, the present work underscores a central role of mitochondria in pathological growth responses in the heart and supports current efforts to design mitochondrial targeted therapies for HF [34,35,36,37]. IF1 appears to be both an attractive and feasible target for pharmacological modulation as several compounds are currently in various stages of clinical development [38,39].

## 4. Material and Methods

The use of animals for these studies was in accordance with the National Institutes of Health *Guide for the Care and Use of Laboratory Animals*. The studies were submitted to and approved by the Institutional Animal Care and Use Committee of the University of Groningen or the University of California San Diego.

### 4.1. Heart Failure Samples

mRNA was isolated from viable left ventricular tissue from three different murine models for chronic heart failure; first, Gαq-40 transgenic mice, which have been described previously [15]. Second, transverse aortic constriction (TAC) was performed with a 7–0 nylon suture between the carotid arteries around a 27G needle, as described [16]. Third, post-myocardial infarction (MI), heart failure was achieved through a permanent ligation of the left coronary artery with a Premilene 6–0 suture [16]. Human myocardial tissue was obtained from patients with end-stage ischemic heart failure (*n* = 10) and from donor hearts rejected due to technical reasons (*n* = 19) within the heart transplant program at the Heart and Diabetes Center NRW. The study was approved by the local ethical committee and conducted in accordance with the guidelines in the Declaration of Helsinki.

### 4.2. Neonatal Rat Cardiomyocyte Isolation

The use of animals for these studies was in accordance with the National Institutes of Health *Guide for the Care and Use of Laboratory Animals*. The study was submitted to and approved by the Committee for Animal Experiments of the University of Groningen. Euthanasia was performed by quick decapitation. Primary neonatal rat ventricular myocytes (NRVMs) were isolated from 1–2-day-old neonatal rats as previously described [40]. NRVMs were grown in MEM (Sigma M1018, St. Louis, MO, USA) supplemented with 5% fetal bovine serum (FBS) (Thermo Fisher SV30160, Waltham, MA, USA) and penicillin–streptomycin (100 U/mL–100 μg/mL) (Thermo Fisher 15070063, Waltham, MA, USA). The IF1 wild-type sequence (IF1-WT) was cloned into adenovirus pSF-Ad5-WT OG617 using the E1A promoter (Oxford Genetics, Oxford, UK). The inactive form of the IF1 (E55A) sequence was cloned previously by Prof. David Sabatini (Addgene #85404, Watertown, MA, USA) [18]. NRVMs were infected with recombinant adenovirus particles (multiplicity of infection (MOI) 50–300) 24 h after isolation and 1% FBS in DMEM with penicillin–streptomycin (100 U/mL–100 μg/mL) the day after.

### 4.3. Western Blot

SDS-PAGE assays were performed by separating 10–20 µg of protein after cell lysis with radioimmunoprecipitation assay (RIPA) buffer plus protease inhibitor (Roche 11873580001, Basel, Switzerland), phosphatase inhibitor cocktail 3 (Sigma P2850, St. Louis, MO, USA), sodium orthovanadate (Sigma S6508, St. Louis, MO, USA), and phenylmethylsulfonyl fluoride (PMSF) (Roche 10837091001, Basel, Switzerland). Protein quantification was measured with a Pierce™ Bicinchoninic acid (BCA) protein assay kit (Thermo Fischer 23225, Waltham, MA, USA). The samples were incubated with 5× loading buffer (final 10% glycerol (Sigma G5516, St. Louis, MO, USA), 2% Sodium Dodecyl Sulphate (SDS) (Sigma 05030, St. Louis, MO, USA), 0.3% Dithiothreitol (DTT) (Sigma 43819, St. Louis, MO, USA), 66 mM Tris (Sigma T6066,St. Louis, MO, USA), and bromophenol blue (Sigma 114391, St. Louis, MO, USA) and boiled at 95 °C. For OXPHOS (Abcam ab110413, Cambridge, UK) detection, the samples were incubated with sample buffer and warmed at 37 °C. After electrophoresis, proteins were transferred to Polyvinylidene fluoride (PVDF) membranes (BioRad 162-0177, Hercules, CA, USA). Membranes were blocked and incubated with primary and secondary HRP-labeled antibodies (Dako, Denmark) before detection with chemiluminescence (ECL) (PerkinElmer NEL112001EA, Waltham, MA, USA). The specific primary and secondary antibodies that were used are depicted in Appendix A.

### 4.4. Real-Time PCR

To analyze gene expression, total RNA was isolated using TRI reagent according to the protocol provided (Sigma T9424, St. Louis, MO, USA). RNA concentrations have been determined with a NanoDrop™ 2000/2000c Spectrophotometer (Thermo Fischer Scientific ND-2000, Waltham, MA, USA). cDNA was synthetized by reverse transcription using QuantiTect Reverse Transcription Kit (Qiagen 205313, Hilden, Germany), and real-time qPCR was performed with IQ SYBR green (Bio-Rad 170-8885, Hercules, CA, USA) using specific primers (see Appendix A). Relative expression levels were calculated using 2^(-ΔΔCT)^.

To assess the mitochondrial DNA to nuclear DNA ratio and DNA damage, total DNA including mtDNA was extracted from the non-infarcted left ventricle using NucleoSpin Tissue XS (Macherey-Nagel 740.901.50, Düren, Germany). The mtDNA-to-nDNA ratio was determined by quantitative real-time polymerase chain reaction (qRT-PCR), as described previously [29]. Mitochondrial DNA copies were corrected for nuclear DNA values, and the calculated values were expressed relative to the control group per experiment. To determine DNA damage (lesions/10 kb), two protocols for RT-QPCR were performed. Firstly, a short mitochondrial DNA fragment was amplified with a short-run protocol. Secondly, a long mitochondrial DNA fragment was amplified with a long-run protocol. The D-loop mitochondrial genomic region was amplified by a semi-long-run qRT-PCR. A list of primer sequences used for this manuscript is listed in the online data supplement (Appendix A).

### 4.5. Mitochondrial Membrane Potential

For mitochondrial potential measurements, the cells were plated in individual Petri dishes with a glass bottom FluoroDish cell culture dish, 35 mm (WPI FD35-100, Sarasota, FL, USA). Tetramethylrhodamine (TMRE; 100 nM; Thermo Fisher T669, Waltham, MA, USA) was added to the culture media and incubated for 30 min at 37 °C, 5% CO_2_. Hereafter, the cells were washed three times with Krebs buffer and maintained in this medium at room temperature for the rest of the experiment. A time-lapse series (200 pictures in 12 min) was made using a Leica AF-6000 epifluorescence microscope. After 1 min of base line measurements, rotenone (4 µM) (Sigma R8875, St. Louis, MO, USA) and antimycin-A (2 µM) (Sigma A8674, St. Louis, MO, USA) were dissolved in Krebs buffer and added to the medium. One minute before the end of the time lapse, FCCP (20 µM) (Enzo BML-CM120-0010, Farmingdale, NY, USA) was added. Imaging analysis was performed using ImageJ (NIH, Bethesda, MD, USA). 

### 4.6. ATP Measurements

ATP concentrations were measured in whole-cell lysates using an ATP Assay Kit (colorimetric/fluorometric) from Abcam (#ab83355, Cambridge, UK) according to the manufacturer’s instructions. Results are normalized for protein concentration.

### 4.7. Seahorse Assays

Measurements of the oxygen consumption rate (OCR) and extracellular acidification rate (ECAR) were performed using the Agilent Seahorse Bioscience XF96 analyzer. Briefly, 80,000 cells/well previously treated with specific adenoviruses were cultured in 96-well culture plates (Agilent 100850-001, Santa Clara, CA, USA) using the XF MitoStress and XF GlycoStress protocol, according to the guidelines provided by the supplier (Agilent Technologies, Santa Clara, CA, USA). Every condition was simultaneously measured in at least eight wells per experiment. Results were normalized to protein levels detected with the BCA method performed at the end of the experiment. Data were obtained using Seahorse Wave desktop software (Agilent Technologies, Santa Clara, CA, USA).

### 4.8. Mitochondrial Morphology

Cells were plated on glass coverslips and stained with MitoTracker Deep Red (200 nM) for 30 min (Invitrogen M22426, Carlsbad, CA, USA). Immediately after mitochondrial staining, cells were washed three times in PBS 1×, fixed with 4% paraformaldehyde (Merck 4005, Darmstadt, Germany), and processed as described in the cell size section. FITC-labeled α-actinin (Sigma A7811, St. Louis, MO, USA) was employed as a marker for cardiomyocytes. Image acquisition was performed using Leica Sp8 Lightning confocal microscope (Leica, Wetzlar, Germany). Z-stacks was obtained from two independent channels, and imaging deconvolution was performed using Scientific Volume Imaging (SVI) (Huygens Pro, The Netherlands). Processed images were analyzed by Imaris software (Oxford Instruments, Abingdon, UK). The cell surface was measured, and 3D reconstruction was performed by surface area at a detailed level (0.2 µM) from red (mitochondria) and green channel (α-actinin), respectively. The number of particles and total mitochondrial volume were quantified per cell.

### 4.9. Mitochondrial Isolation

NRVMs were rinsed with PBS 1× and 1 mL of mitochondrial isolation buffer (sucrose 70 mM, mannitol 220 mM, KH_2_PO_4_ 10 mM, MgCl_2_ 5 mM, HEPES 2 mM, EGTA 1 mM, and BSA 0.2%) were added. Immediately, the NRVMs were scraped and homogenized with a glass rod. After, the homogenized cells were centrifuged at 150× *g* for 10 min at 4 °C. Supernatants were collected, and a second spin was performed at 12,000× *g* for 15 min at 4 °C. The cytosolic fraction was gathered from the supernatant, and the mitochondrial fraction was resuspended in RIPA buffer. Both fractions were treated with protease (Roche 11873580001, Basel, Switzerland) and phosphatase inhibitors (Sigma P2850, St. Louis, MO, USA). The BCA assay was used for protein quantification (Thermo Fischer 23225, Waltham, MA, USA).

### 4.10. Mitochondrial and Cytosolic Ca^2+^ Levels

For mitochondrial Ca^2+^ content/release, the cells were plated in individual Petri dishes with a glass bottom: FluoroDish cell culture dish, 35 mm (WPI FD35-100, Sarasota, FL, USA). Two Ca^2+^ labels were used for the measurement of mitochondrial Ca^2+^ content: Rhod2-AM and Fluo4-AM (5 µM; Thermo Fisher F14201, Waltham, MA, USA). Thapsigargin (10 µM; Enzo BML-PE180-001, New York, NY, USA) was added to the culture media and incubated for 30 min at 37 °C, 5% CO_2_. Hereafter, the cells were washed three times with Ca^2+^-free Krebs buffer. A time lapse (200 pictures in 30 min) was made using a Leica AF-6000 microscope (Leica, Wetzlar, Germany). After 2 min, FCCP (20 µM) suspended in Ca^2+^-free Krebs buffer was added. Imaging analysis was performed using ImageJ (NIH, Bethesda, MD, USA).

For cytosolic Ca^2+^ measurement, the cells were plated in individual Petri dishes with glass bottoms (35 mm). FURA 2-AM (4 µM; Abcam ab120873, Cambridge, UK) was added to the culture medium and incubated for 45 min at 37 °C, 5% CO_2_. Hereafter, the cells were washed three times with Ca^2+^-free Krebs buffer. A time lapse of 5 min was measured with 340 and 380 nm wavelength using Leica DM IRE2 and Polychorome V (Till photonics, Graefelfing, Germany). After 1 min, KCL (50 mM) suspended in Ca^2+^-free Krebs buffer was added. The ratio of 340/380 wavelength fluorescence analysis was performed using ImageJ (NIH, Bethesda, MD, USA). 

### 4.11. Cell Size

For cell-size measurement, cells were cultured on laminin (Millipore CC095)-coated coverslips for 48 h, and then transfected with the specific adenoviruses. The cardiomyocytes were fixed with 4% paraformaldehyde (Merck 4005, Darmstadt, Germany) in a phosphate buffer for 5 min at room temperature. After, the cells were washed in PBS 1×, followed by permeabilization with PBS + 0.3% Triton-X100 (SigmaT9284, St. Louis, MO, USA) on ice for 5 min. For image acquisition, we used Leica SP8 epifluorescence microscopy (Leica, Wetzlar, Germany), and for the determination of cellular area analysis, we used ImageJ software (NIH, Bethesda, MD, USA). Five observation fields were selected randomly on each cover slip, and 5–10 cells within each observation field were selected for the determination of the mean cardiomyocyte surface area according to the image analysis system. For the measurements, we used at least five different fields from five independent cultures in each condition (>50 cells).

### 4.12. Statistical Analysis

All data are presented as mean ± SEM. Comparisons between groups were performed using the Student *t* test, the Mann–Whitney *U* test, Kruskal–Wallis test, or one-way ANOVA, followed by the Tukey post hoc test, where appropriate. A *p* value < 0.05 was considered statistically significant.

## 5. Conclusions

IF1 represents a novel member of the fetal-like gene program that contributes to mitochondrial dysfunction and pathological cardiac remodeling in HF. Furthermore, we present evidence for a novel, ATP-synthase-independent, role for IF1 in mitochondrial Ca^2+^ handling and mitochondrial-to-nuclear crosstalk involving CaMKII.

## Figures and Tables

**Figure 1 ijms-22-04427-f001:**
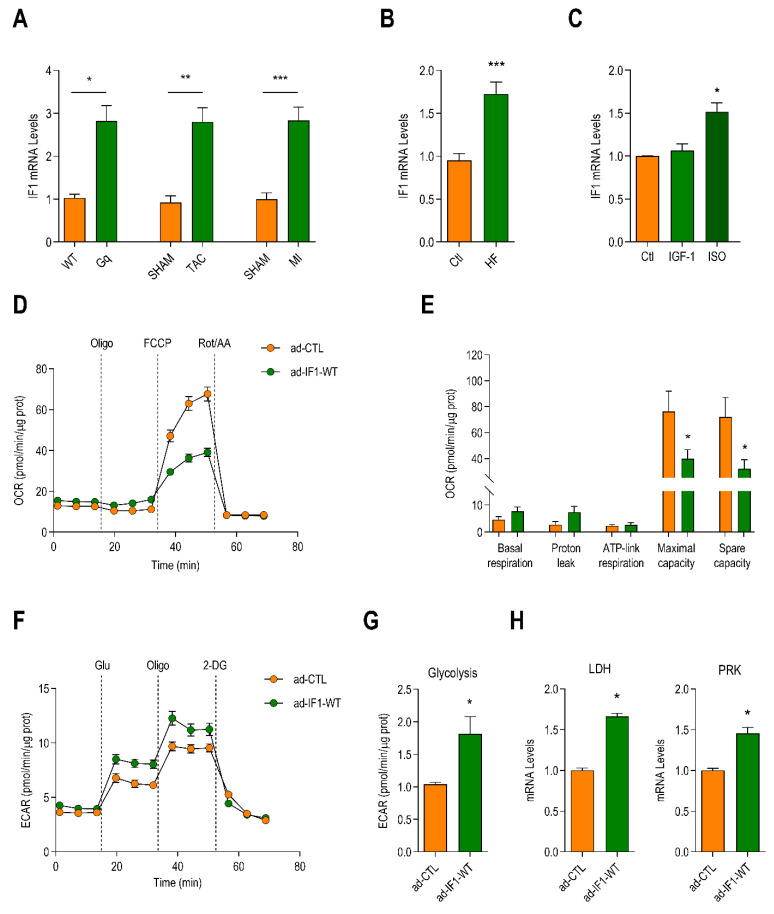
Expression of ATPase inhibitory factor-1 (IF1) is increased in heart failure, decreases mitochondrial respiratory capacity, and promotes glycolysis in cardiomyocytes. (**A**) IF1 mRNA levels in mice with heart failure induced by cardiomyocyte autonomous expression of Gαq and wild-type (WT) littermate controls (*n* = 4), and mice with heart failure induced by transverse aortic constriction (TAC) (*n* = 6) or myocardial infarction (MI) (*n* = 6) surgeries and sham-operated controls (SHAM). (**B**) mRNA expression of IF1 in cardiac lysates from patients with end-stage heart failure and normal control hearts (*n* = 11). (**C**) IF1 mRNA expression in neonatal rat ventricular myocytes (NRVMs) treated with insulin-like growth factor-1 (IGF-1, 10 nM), isoproterenol (ISO, 100 nM) or vehicle (Ctl) for 48 h (*n* = 4). (**D**) Neonatal rat ventricular myocytes (NRVMs) were infected with an adenoviral vector expressing human IF1 (ad-IF1-WT) or a control virus expressing green fluorescent protein (GFP) (ad-CTL) for 48 h. The line graph depicts changes in the oxygen consumption rate (OCR) of NRVMs after serial treatments with oligomycin (oligo), FCCP, and rot + AA assessed with the Seahorse system. The graph represents five independent experiments. (**E**) Bar graph depicting differences in basal respiration, ATP-linked respiration, maximal respiration, and mitochondrial spare capacity (*n* = 5). Data are presented as mean ± SEM. * *p* < 0.05 and ** *p* < 0.01 by parametric test *t*-test. (**F**) Changes in extracellular acidification rate (ECAR) in cells infected with ad-IF1-WT or ad-CTL after serial additions of glucose, oligomycin, and 2-deoxyglucose (2-DG, *n* = 4). (**G**) Bar graph depicting differences in glycolysis in NRVMs treated as in F (*n* = 4). (**H**) mRNA levels of lactate dehydrogenase (LDH) and pyruvate kinase (PRK) assayed with RT-qPCR (*n* = 4). Data are presented as mean ± SEM. * *p* < 0.05, ** *p* < 0.01, and *** *p* < 0.001 vs. SHAM/Ctrl/ad-CTL using the Mann–Whitney *U* test or *T*-test where appropriate.

**Figure 2 ijms-22-04427-f002:**
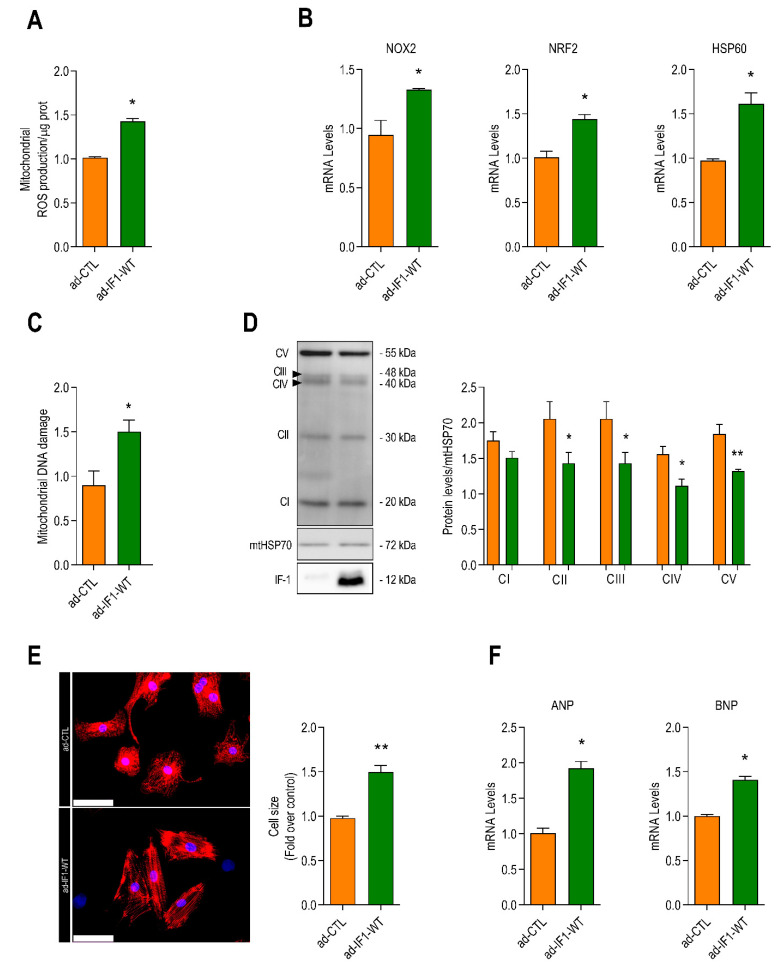
IF1 overexpression promotes mitochondrial reactive oxygen species (ROS) production and cardiomyocyte hypertrophy. NRVMs were infected with an adenoviral vector expressing the human IF1 (ad-IF1-WT) or a control virus (ad-CTL) for 48 h. All bar graphs depict the results of at least three independent experiments. (**A**) Basal mitochondrial ROS emissions detected with the mitochondrial ROS sensor MitoSOX^®^ (*n* = 4). (**B**) mRNA expression of the mitochondrial ROS-sensitive genes NADPH oxidase 2 (*NOX2*), nuclear receptor factor-2 (*NRF2*), and heat shock protein-60 (*HSP60*) (*n* = 4). (**C**) Semi-log-run polymerase chain reaction was performed using specific primers for mitochondrial DNA fragments in the mitochondrial D-Loop region to determine oxidative damage to mitochondrial DNA (*n* = 4). (**D**) Changes in the protein levels of mitochondrial respiratory chain complexes in cells infected with ad-IF1-WT and ad-CTL detected by Western blot. Different complex levels were normalized for mitochondrial heat shock protein-70 (mtHSP70) levels (*n* = 4). (**E**) Typical example of immunofluorescent staining of NRVMs cultured with a fluorescent-labeled anti-α-actinin (red, left panel) and (right panel) average cardiomyocyte surface area of α-actinin-positive cells (*n* = 5). (**F**) mRNA expression of atrial natriuretic peptide (*ANP*) and B-type natriuretic peptide (*BNP*). Data are presented as mean ± SEM. * *p* < 0.05 and ** *p* < 0.01 vs. ad-CTL by nonparametric Mann–Whitney *U* test using the Mann–Whitney *U* test or *T*-test where appropriate.

**Figure 3 ijms-22-04427-f003:**
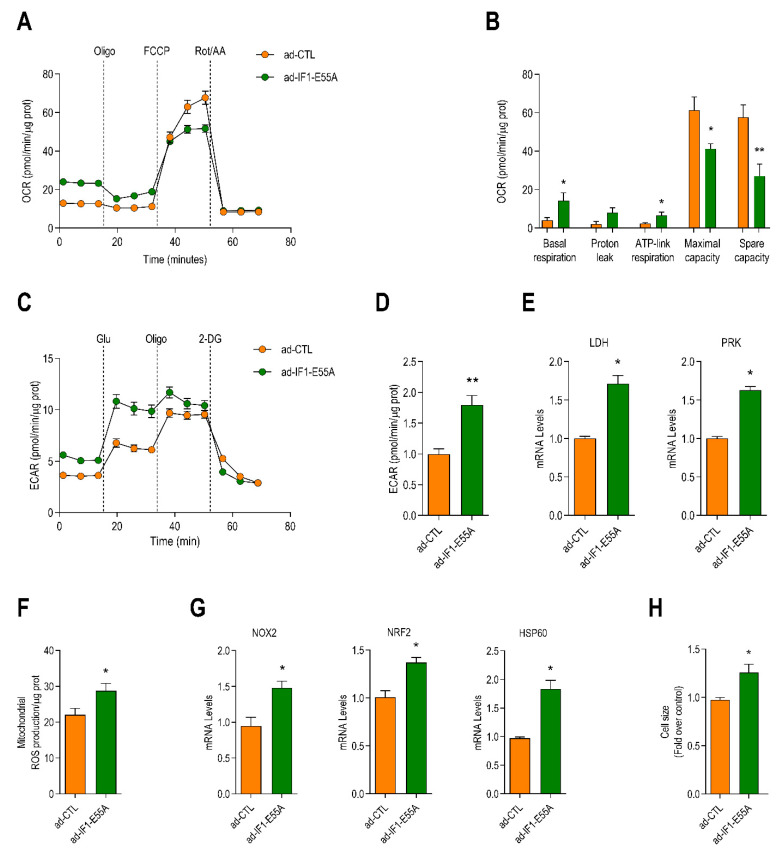
An IF1 mutant incapable of binding to ATP synthase induces similar degrees of mitochondrial stress and cardiomyocyte hypertrophy. NRVMs were infected with an adenovirus expressing an IF1 mutant harboring an E55A substitution that renders the protein unable to interact with ATP synthase or control virus (ad-CTL) for 48 h. (**A**) Changes in the oxygen consumption rate (OCR) of NRVMs after serial treatments with oligomycin (oligo), FCCP, and rot + AA assessed with the Seahorse system (*n* = 5). (**B**) Bar graph depicting differences in basal respiration, ATP-linked respiration, maximal respiration, and mitochondrial spare capacity (*n* = 5). (**C**) Changes in the extracellular acidification rate (ECAR) in cells infected with ad-IF1-E55A or ad-CTL after serial additions of glucose, oligomycin, and 2-deoxyglucose (2-DG, *n* = 4). (**D**) Bar graph depicting differences in glycolysis in NRVMs treated as in F (*n* = 4). (**E**) mRNA levels of lactate dehydrogenase (*LDH*) and pyruvate kinase (*PRK*). (**F**) Basal mitochondrial ROS levels assayed with MitoSOX^®^ (*n* = 4). (**G**) mRNA levels of *NOX2*, *NRF2*, and HSP60. (**H**) Bar graph depicting differences in cardiomyocyte cross-sectional area (*n* = 5). Data are presented as mean ± SEM. * *p* < 0.05 and ** *p* < 0.01 vs. ad-CTL using the Mann–Whitney *U* test or *T*-test where appropriate.

**Figure 4 ijms-22-04427-f004:**
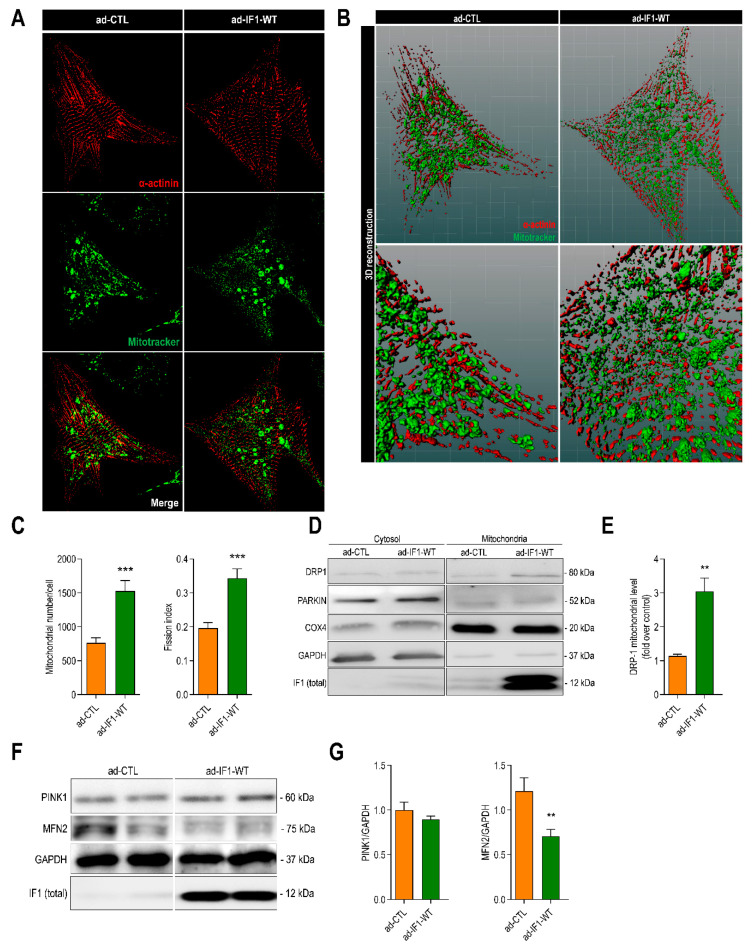
IF1 induces mitochondrial fission by promoting dynamin-related protein 1 (DRP1) translocation. NRVMs were infected with adenovirus ad-IF1-WT or empty vector (ad-CTL) for 48 h. (**A**) Representative Z-stack image of neonatal rat ventricular myocyte stained with MitoTracker^®^ Deep Red (green) and anti-α-actinin (red) using confocal microscopy. (**B**) 3D imaging reconstruction was employed for mitochondrial network analysis (30–36 Z-stacks slices were recorded per cell). Representative example of confocal stack acquisition followed by 3D reconstruction after image deconvolution of cells stained as in A. (**C**) Bar graphs depicting differences in mitochondrial number per cell (left bar graph) and fission index (right bar graph). Both ratios were calculated from 3D multislice reconstruction values. Fission index estimation was measured using the number of mitochondrial normalized for mitochondrial volume per cell (ad-CTL, *n* = 17 cells and ad-IF1-WT, *n* = 18 cells). (**D**) Dynamin-related protein 1 (DRP1), cytochrome c oxidase subunit 4 (COXIV), and glyceraldehyde 3-phosphate dehydrogenase (GAPDH) protein level in mitochondrial and cytosolic fractions of NRVMs infected with IF1-WT and ad-CTL and (left panel). (**E**) DRP1 protein levels normalized for COXIV in the mitochondrial fraction (*n* = 4). (**F**) PTEN-induced kinase 1 (PINK1) and mitofusin-2 (MFN2) protein levels normalized for GAPDH. (**G**) Bar graphs depict densitometric analysis from protein levels normalized using immunoblot. Data are presented as mean ± SEM. ** *p* < 0.01, and *** *p* < 0.001 vs. ad-CTL using the Mann–Whitney *U* test or *T*-test where appropriate.

**Figure 5 ijms-22-04427-f005:**
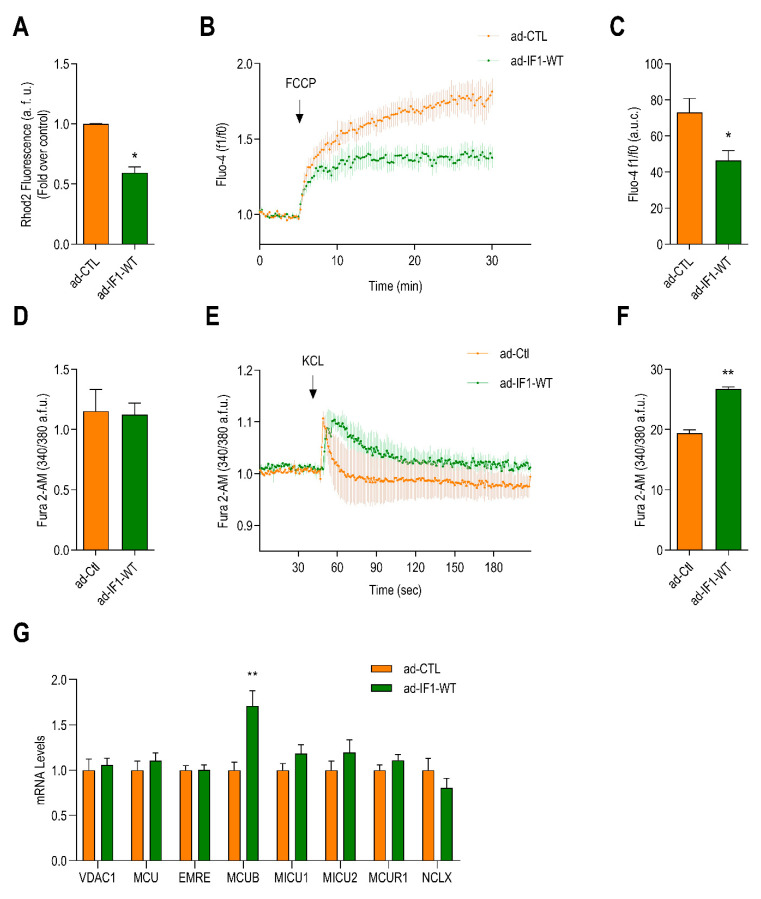
IF1 reduces mitochondrial calcium, promotes sarcoplasmic reticulum (SR) calcium overload, and activates calcium–calmodulin kinase II (CAMKII) signaling. NRVMs were infected with adenovirus ad-IF1-WT and ad-CTL for 48 h. (**A**) Bar graph depicting differences in basal mitochondrial Ca^2+^ detected by with Rhod-2 AM (*n* = 4). (**B**) Time lapse of mitochondrial Ca^2+^ release measurements using Fluo-4 AM before and after addition of FCCP (*n* = 5). (**C**) Differences in the area under the curve of Fluo-4 AM fluorescence after the addition of FCCP (*n* = 5). (**D**) Cytosolic Ca^2+^ measurements using Fura-2 AM in cells infected with ad-IF1-WT or ad-CTL using an epifluorescence microscope (*n* = 4). (**E**) Representative experiment of potassium chloride (KCL)-induced sarcoplasmic reticulum (SR) Ca^2+^ release (*n* = 3). (**F**) Bar graph depicting differences in the area under the curve of Fura-2 AM 340/380 fluorescence ratio after the addition of KCL (*n* = 3). (**G**) Bar graphs depicting mRNA levels of different calcium regulatory genes assessed by qPCR; voltage-dependent anion channel 1 (*VDAC1*); mitochondrial calcium uniporter (*MCU*); essential MCU regulatory element (*EMRE*),; dominant-negative pore-forming subunit of the MCU ß (*MCUB*); mitochondrial calcium uptake protein 1 and 2 (*MICU1* and *MICU2* respectively); MCU regulator 1 (*MCUR1*); mitochondrial sodium calcium exchanger (*NCLX*), (*n* = 4). Data are presented as mean ± SEM. * *p* < 0.05 and ** *p* < 0.01 vs. ad-CTL using the Mann–Whitney *U* test or *T*-test where appropriate.

**Figure 6 ijms-22-04427-f006:**
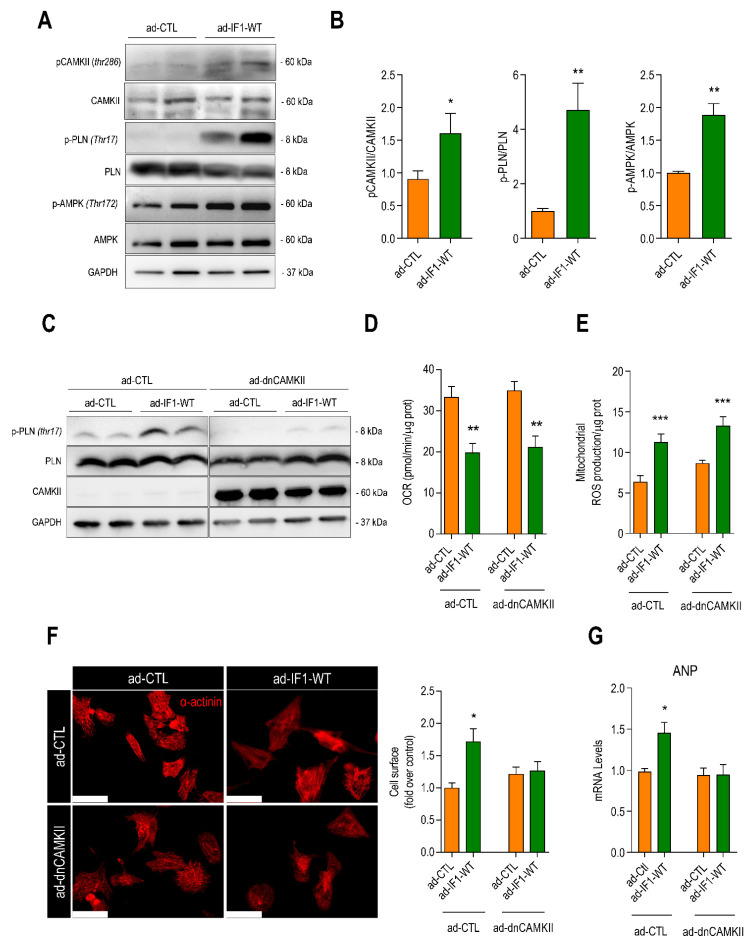
IF1-induced cardiomyocyte hypertrophy is dependent upon CaMKIIδ activation. NRVMs were infected with adenovirus ad-IF1-WT and ad-CTL for 48 h. (**A**) Representative immunoblot depicting phosphorylation of CAMKII (threonine 286) and CaMKII-dependent phosphorylation of phospholamban (p-PLN, thr17) and adenosine monophosphate-activated protein kinase (p-AMPK, thr172), normalized for the total protein (left panel). (**B**) Bar graphs depicting differences in phosphorylation between experimental groups (*n* = 7). (**C**) NRVMs were co-infected with adenovirus expressing IF1-WT or ad-CTL in the presence or absence of a catalytically dead mutant of CAMKIIδC (ad-dnCAMKII) for 48 h. Representative immunoblots from whole-cell lysate using an antibody specific for p-PLN (thr17), total PLN, CAMKII (PAN), and GAPDH (*n* = 4). (**D**) Maximal mitochondria respiration through oxygen consumption rate (OCR) recorded using Mito Stress Test Seahorse assay. (**E**) Bar graph depicting basal mitochondrial ROS levels assayed with MitoSOX^®^ (*n* = 4). (**F**) Typical example of immunofluorescent staining of cultured NRVMs with a fluorescent-labeled anti-α-actinin (red, left panel) and (right panel) average cardiomyocyte surface area of α-actinin-positive cells (scale bar represents 50 µm, *n* = 4). (**G**) *ANP* mRNA level normalized for *36B4* (*n* = 4). * Data are presented as mean ± SEM. * *p* < 0.05, ** *p* < 0.01, and *** *p* < 0.001 vs. ad-CTL using the Mann–Whitney *U* test (**A**,**B**) or the Kruskal–Wallis test (**D**–**G**).

**Figure 7 ijms-22-04427-f007:**
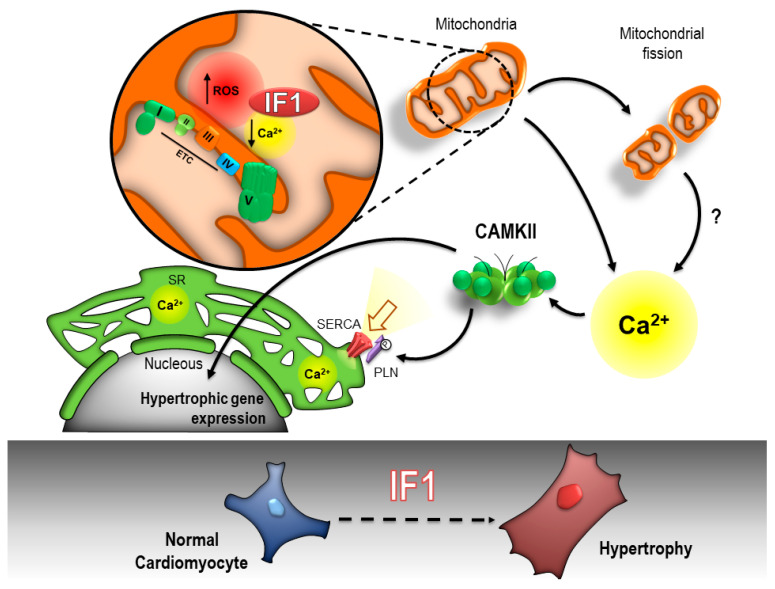
Schematic representation of the proposed mechanism responsible for IF1-mediated cardiomyocyte hypertrophy. Under conditions of respiratory collapse, ATPase inhibitory factor-1 (IF1) prevents maladaptive ATP hydrolysis by ATP synthase (complex V). The function of IF1 under normal respiring conditions remains poorly described. We present evidence for a novel, ATP-synthase-independent, role for IF1 in mitochondrial calcium handling and mitochondrial-to-nuclear crosstalk involving calcium–calmodulin kinase II (CaMKII). Specifically, we found that IF1 induced mitochondrial oxidative stress, promoted mitochondrial fission, and compromised mitochondrial calcium handling in primary cardiomyocytes. These perturbations resulted in sarcoplasmic reticulum calcium overload and the activation of CaMKII-mediated pathological cardiomyocyte hypertrophy. ETC, electron transport chain, ROS, reactive oxygen species, SERCA, sarcoplasmic/endoplasmic reticulum calcium ATPase; PLN, phospholamban; SR, sarcoplasmic reticulum.

## Data Availability

Data and materials will be available to third parties at a reasonable request.

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
