# Peer review of "ATPase Inhibitory Factor-1 Disrupts Mitochondrial Ca2+ Handling and Promotes Pathological Cardiac Hypertrophy through CaMKIIδ"

_ijms, 2021, doi:10.3390/ijms22094427_

Round 1
Reviewer 1 Report
The authors of the article present a very complete elaborate both in the
experimental form and in the description of the experiments carried out,
although in some sections the information is many and can lead to confusion.
I believe that although the article is complete and well structured, it
nevertheless needs a lightening of the form in order to make it easier to
understand. Therefore I invite the authors to streamline the part relating
to the description of the experiments especially in the part relating to
the results e description. The figures are good and have a very good resolution.
After these considerations the paper can be accepted for pubblication.
Author Response
The authors present a very elaborative study. I believe that although the article is complete and well structured, it nevertheless needs a lightening of the form in order to make it easier to
understand. Therefore I invite the authors to streamline the part relating to the description of the experiments especially in the part relating to the results e description.
We thank the reviewer for noticing the quality of our work and agree that a lightening of the form could improve the clarity of the results. We have therefore made an extensive adjustment to the whole manuscript, with emphasis to the results section.
Reviewer 2 Report
In the paper of Pavel-Giani et al, they investigated whether ATPase inhibitor factor-1 (IF1) promotes mitochondrial dysfunction and pathological cardiomyocyte hypertrophy in the context of heart failure (HF). They found that IF1 represents a novel ATP-synthase independant, role of IF1 in mitochondrial calcium handling and mitochondrial-to-nuclear crosstalk involving CaMKII. The paper is well written and the methodology is well performed. However I have still some points to discuss.
1) The evaluation of the Nrf2 pathway is limited (only mRNA expression). When activated, Nrf2 is translocated to the nucleus where it induces the expression of the antioxidative defense. The authors should show this nuclear translocation.
2) If mitochondrial ROS plays an important role in IF1 expression, does it mean that the addition of an mitochondrial specific antioxidant such mito-TEMPO could prevent the hyperthophy of cardiomyocytes?
3) What is the clinical relevance of these findings? Please discuss.
Author Response
We thank the reviewer for considering our paper to be well written and well performed. The reviewer has 3 points that could improve the manuscript.
- The reviewer argues that the evaluation of the Nrf2 pathway is limited. As rightly noticed by the reviewer, Nrf2 is a transcription factor that is translocated to the nucleus to activate antioxidant response elements (ARE). The author suggest that we should determine to what extent IF1 induces nuclear translocation of Nrf2.
We acknowledge that the effect of IF1 on the NRF2 pathway is limited and that measuring mRNA expression does not recapitulate the biological function of NRF2 to its full extent. However, the mRNA expression of NRF2 is controlled by the very ARE that is activated by NRF2, making NRF2 mRNA levels a reasonable marker for both oxidative stress and the transcriptional activity of NRF2.[1,2] As stated, we employed NRF2, along with NOX-2 and HSP60, as molecular markers for oxidative stress and do not consider these factors to be the primary mechanisms responsible IF1-induced cardiomyocyte remodeling. While of interest, an in-dept investigations into nuclear translocation of Nrf2 is therefore beyond the scope of the current investigation.
- If mitochondrial ROS plays an important role in IF1 expression, does in mean that mito-TEMPO can prevent hypertrophy in cardiomyocytes?
The reviewer rightly acknowledges that IF1-induced oxidative stress could represent the dominant upstream factor that causes mitochondrial fission, disruption of mitochondrial calcium handling and subsequent activation of CaMKII. If so, a mitochondrial antioxidant would suffice to prevent IF1-induced cardiomyocyte hypertrophy.
While we agree with the reviewer that this would be an interesting area of investigation, it will be very difficult to untangle the exact role of each downstream effect of IF1. For instance, in addition to Ca2+, CaMKII is also activated by ROS and CaMKII promotes mitochondrial ROS emissions by itself.[3,4] Merely adding an antioxidant would not provide definitive proof as to whether increased ROS is an upstream or a downstream regulator of cardiomyocyte hypertrophy. To address this limitation in our paper we have changed the limitations section of the manuscript on page 17 into:
Another limitation of our study is that we do not provide mechanisms responsible for IF1-induced mitochondrial oxidative stress. In addition, we do not provide mechanistic insights into the role of mitochondrial oxidative stress in cardiomyocyte hypertrophy downstream of IF1. In our opinion, this is beyond the scope of the current investigation and future studies are required to address these questions.
- What are the clinical implications of these findings? Please discuss.
We agree with the reviewer that the clinical implications of our study are only briefly described and in generic terms. We have therefore added the following to the discussion on page 18 from line 1107 onwards:
IF1 appears to be both an attractive and feasible target for pharmacological modulation as several compounds are currently in various stages of clinical development.[5,6]
- Vashi, R.; Patel, B.M. NRF2 in Cardiovascular Diseases: a Ray of Hope! J Cardiovasc Transl Res 2020, doi:10.1007/s12265-020-10083-8.
- Kwak, M.K.; Itoh, K.; Yamamoto, M.; Sutter, T.R.; Kensler, T.W. Role of transcription factor Nrf2 in the induction of hepatic phase 2 and antioxidative enzymes in vivo by the cancer chemoprotective agent, 3H-1, 2-dimethiole-3-thione. Mol Med 2001, 7, 135-145.
- Westenbrink, B.D.; Ling, H.; Divakaruni, A.S.; Gray, C.B.; Zambon, A.C.; Dalton, N.D.; Peterson, K.L.; Gu, Y.; Matkovich, S.J.; Murphy, A.N.; et al. Mitochondrial reprogramming induced by CaMKIIdelta mediates hypertrophy decompensation. Circ Res 2015, 116, e28-39, doi:10.1161/CIRCRESAHA.116.304682.
- Westenbrink, B.D.; Edwards, A.G.; McCulloch, A.D.; Brown, J.H. The promise of CaMKII inhibition for heart disease: preventing heart failure and arrhythmias. Expert Opin Ther Targets 2013, 17, 889-903, doi:10.1517/14728222.2013.809064.
- Ivanes, F.; Faccenda, D.; Gatliff, J.; Ahmed, A.A.; Cocco, S.; Cheng, C.H.; Allan, E.; Russell, C.; Duchen, M.R.; Campanella, M. The compound BTB06584 is an IF1 -dependent selective inhibitor of the mitochondrial F1 Fo-ATPase. Br J Pharmacol 2014, 171, 4193-4206, doi:10.1111/bph.12638.
- Strobbe, D.; Pecorari, R.; Conte, O.; Minutolo, A.; Hendriks, C.M.M.; Wiezorek, S.; Faccenda, D.; Abeti, R.; Montesano, C.; Bolm, C.; et al. NH-sulfoximine: A novel pharmacological inhibitor of the mitochondrial F1 Fo -ATPase, which suppresses viability of cancerous cells. Br J Pharmacol 2021, 178, 298-311, doi:10.1111/bph.15279.